# Mimicking Sampson’s Retrograde Menstrual Theory in Rats: A New Rat Model for Ongoing Endometriosis-Associated Pain

**DOI:** 10.3390/ijms21072326

**Published:** 2020-03-27

**Authors:** Eleonora Persoons, Katrien De Clercq, Charlotte Van den Eynde, Sílvia João Poseiro coutinho Pinto, Katrien Luyten, Rita Van Bree, Carla Tomassetti, Thomas Voets, Joris Vriens

**Affiliations:** 1Laboratory of Endometrium, Endometriosis & Reproductive Medicine, Department of Development and Regeneration, KU Leuven, Herestraat 49 box 611, 3000 Leuven, Belgium; eleonora.persoons@kuleuven.be (E.P.); katrien.declercq@kuleuven.be (K.D.C.); charlotte.vandeneynde@kuleuven.be (C.V.d.E.); katrien.luyten@kuleuven.be (K.L.); rita.vanbree@kuleuven.be (R.V.B.); carla.tomassetti@uzleuven.be (C.T.); 2Laboratory of Ion Channel Research, Department of Cellular and Molecular Medicine, KU Leuven, VIB Center for Brain & Disease Research, Herestraat 49 box 802, 3000 Leuven, Belgium; silvia.pinto@kuleuven.vib.be (S.J.P.c.P.); thomas.voets@kuleuven.vib.be (T.V.); 3Leuven University Fertility Center, University Hospitals Leuven, Herestraat 49, 3000 Leuven, Belgium

**Keywords:** endometriosis, menstrual tissue, menstruating rat, ongoing pain

## Abstract

Endometriosis is a prevalent gynecologic disease, defined by dysfunctional endometrium-like lesions outside of the uterine cavity. These lesions are presumably established via retrograde menstruation, i.e., endometrial tissue that flows backwards during menses into the abdomen and deposits on the organs. As ongoing pain is one of the main pain symptoms of patients, an animal model that illuminates this problem is highly anticipated. In the present study, we developed and validated a rat model for ongoing endometriosis-associated pain. First, menstrual endometrial tissue was successfully generated in donor rats, as validated by gross examination, histology and qPCR. Next, endometriosis was induced in recipient animals by intraperitoneal injection of menstrual tissue. This resulted in neuro-angiogenesis as well as established endometriosis lesions, which were similar to their human counterparts, since epithelial and stromal cells were observed. Furthermore, significant differences were noted between control and endometriosis animals concerning bodyweight and posture changes, indicating the presence of ongoing pain in animals with endometriosis. In summary, a rat model for endometriosis was established that reliably mimics the human pathophysiology of endometriosis and in which signs of ongoing pain were detected, thus providing a new research tool for therapy development.

## 1. Introduction

Approximately 10% of reproductive age women, i.e., 176 million women worldwide, suffers from endometriosis [1]. This gynecologic condition is defined by dysfunctional endometrium-like lesions outside of the uterine cavity, and manifests itself in symptoms such as infertility and/or pelvic pain. Despite its high prevalence and burden, the etiology of the disease has been elusive for decades. In the beginning of the 20th century, Sampson postulated several theories of endometriosis development, such as vascular dissemination, celomic metaplasia and retrograde menstruation [2,3]. The latter is the most likely to explain the location of the majority of cases of intra-abdominal endometriosis [2,3]. Retrograde menstruation stipulates that lesions arise from the blood and cells that are shed during menstruation, and are deposited in the abdomen via the fallopian tubes due to a backward menstrual flow [2]. Nonetheless, this process is regarded as non-pathological, as it occurs in 90% of menstruating women, and most women are able to clear this tissue over time [4]. However, in women with endometriosis, these menstrual endometrial cells are able to survive and flourish in the abdominal cavity, presumably by acquiring additional capacities, such as migration, adhesion, proliferation, immune evasion and triggering neuro-angiogenesis, ultimately culminating in endometriosis lesions [5].

Patients suffering from endometriosis consult medical care for ongoing pain symptoms including chronic pelvic pain, dysmenorrhea and dyspareunia. Unfortunately, the treatment options for endometriosis-associated pain are currently limited to a combination of surgery, hormone treatment and regular pain medication. Due to the unavailability of endometriosis-targeted pain therapy, the untreated, persistent pain could potentially modify signal transmission in the central nervous system resulting in chronic pain [6,7]. This process of central sensitization occurs after an undefined period of time, but is not described in all patients [6].

In light of this, global efforts have been made to elucidate the pain pathogenesis of endometriosis, via both clinical and in vivo studies [8]. As clinical research into this topic opposes many ethical hurdles, scientific researchers often prefer animal models to study the etiology and pain pathophysiology of endometriosis. Currently, literature describes several animal models, of which the majority are rodent models. Rodents are easy to handle, time- and cost-effective and can be genetically modified in the context of a research hypothesis. The downside, however, is their estrus cycle, and consequently, the lack of spontaneous decidualization and menstruation, which implies that rodents do not develop endometriosis. To overcome this impediment, the transplantation of endometrial tissue in the rodent’s abdominal cavity was introduced. In this regard, Greaves et al. developed the menstruating mouse model to induce endometriosis, in which menstrual endometrial tissue from donor mice was injected in the peritoneal cavity of recipient mice [9].

Although the model of Greaves et al. is effective, mimics the human pathophysiology and provided more insight in the underlying processes, it has only been used to investigate central sensitization [10], and thus, fails to acknowledge the ongoing endometriosis-associated pain symptoms. As ongoing pain is a prominent feature of the disease, it is of the utmost importance to develop research tools and models to investigate the pathophysiology of ongoing endometriosis-associated pain. Most pain behavior assays for ongoing pain are optimized for larger rodents such as rats. However, to date, no rat model is available that mimics the retrograde menstruation, cf. the menstruating mouse model. On this account, we aimed to develop a novel rat model for endometriosis, in which donor endometrial menstrual tissue is generated to induce endometriosis in recipient rats, and tested them for ongoing endometriosis-associated pain.

## 2. Results

### 2.1. Generation of Menstrual Endometrial Tissue in Donor Rats

The menstruating mouse model of Cousins et al. [11] is able to generate menstrual endometrial tissue by intervening in the normal estrus cycle. In order to mimic the menstruating model in rats, an adaptation of the model was needed. After an optimization process of the high and low estrogen (E2) injection (Appendix A), the following protocol was established (Figure 1). In ovariectomized rats, high estrogen injections (500 ng/100 µL) were dosed to mimic the estrogen-surge before ovulation. Next, the window of implantation was replicated by administering low estrogen injections (5 ng/100 µL) supplemented with a progesterone pellet. As decidualization in rodents solely occurs in response to implanting embryos during the window of implantation, a mechanical substitute was given, i.e., the luminal endometrial surface was scraped while co-injecting oil in the lumen at day 15 of the protocol. The progesterone withdrawal at day 19 of the protocol resulted in the arrangement of menstrual endometrial tissue in rats, i.e., the menstruating rat model (MRM) (Figure 1A).

The decidual character of the uterine horns was validated using gross examination and histology. Note that the stimulation of the uterine horn (i.e., injecting oil into the lumen of the uterine horn at day 15) can have varying results in terms of the decidualization grade at the end of the protocol. By day 19 of the protocol, the stimulated uterine horns were enlarged (Figure 1I), which resulted in a significant increase in uterine weight compared to the paired control horn (605.4 ± 39.7 mg vs. 98.3 ± 5.6 mg (mean ± SEM)) (Figure 1B). Due to stromal tissue expansion, the surface area of the uterine horn increased up to 10-fold compared to the control horn (9.2 ± 0.9 mm² vs. 0.8 ± 0.1 mm² (mean ± SEM)) (Figure 1C) and resulted in the closure of the uterus’ lumen (Figure 1D,E) [12]. In addition, the stromal compartment of the treated horn showed clear positivity for the decidual marker desmin, as opposed to the control horn (85.8 ± 15.9 % vs. 18.2 ± 3.6 % of the stroma (mean ± SEM; *n* = 5)) (Figure 1F,G). Full length or partial decidualization was observed in 61.9% and 22.7% of the stimulated horns, respectively. Only 15.4% of the stimulated horns showed no signs of decidualization at the end of the protocol (Figure 1H).

In order to validate the menstrual characteristics of the tissue, the gene expression profile was investigated using quantitative RT-PCR (RT-qPCR). RNA was isolated from the uterine tissue at day 19 of the protocol, which was subjected to MRM without decidualization stimulus (non-stimulated), MRM without removal of the P4 pellet (decidualized, DECI) or MRM after P4 withdrawal (menstrual, MENS) (Figure 2). Compared to the non-stimulated group, only a significant increase in *prostaglandin-endoperoxide synthase*
*(Ptgs) 2* and *Matrix metalloprotease (Mmp) 3* and *10* was observed in the DECI group. The other genes involved in the hallmarks of menstruation, i.e., vasomodulation, tissue breakdown and inflammation were not upregulated in the decidual tissue. This is in contrast to the MENS group, where a significant upregulation of *Ptgs2*, *Mmp 1*, *3* and *10*, and *interleukin 6 (Il6)* was observed validating the menstrual character of the tissue.

Interestingly, some markers were significantly different between the non-stimulated and DECI group, suggesting their regulation during decidualization. However, the expression of others was not dissimilar between the decidualized endometrium and its internal control (e.g., *Vegfa*). Therefore, it was further investigated whether expression of the latter was due to the MRM, by comparing the non-stimulated group to naïve endometrium collected during a normal estrous cycle (EST). In order to detect whether these genes were expressed because of progesterone, the naïve tissue was isolated from the pro-estrous phase (Appendix A). However, no significant differences were observed between both groups, implying constitutive expression of these markers in the endometrium.

### 2.2. Endometriosis Lesions can be Retrieved from Recipient Rats

Endometriosis was induced in recipient animals by a single intraperitoneal (i.p.) injection of approximately 400 mg menstrual endometrial tissue. To reduce variability, the recipient animals were given high estrogen injections prior to the induction in order to synchronize the estrus cycle (Figure 3A). The inoculation resulted in adhered macroscopically visible endometriosis lesions in 45% of the treated animals after 12 weeks (Figure 3B). However, in 22.5% of the remaining recipient rats, non-adhered tissue could be observed in the abdominal cavity (Figure 3B,F). Similar to their human counterparts, the adhered endometriosis lesions varied in color, abdominal location and size (Figure 3C–F). The majority of the lesions, noticeably, tended to form onto connective tissue, such as those surrounding the stomach and pancreas (64%, 16 out of 25), intestines (12%; three out of 25) and gonadal adipose tissue (12%; three out of 25) (Appendix A). In rare cases, lesions were observed subcutaneously or on the peritoneum itself (8% and 4%, respectively). Furthermore, adhered lesions varied in size from approximately 1 mm^2^ to 25.1 mm^2^, with an average size of 7.4 ± 1.3 mm^2^ (Appendix A). In the majority of animals, the number of adhered endometriosis lesions that could be retrieved per recipient animal was limited to one. However, in some animals, two to four ENDO lesions were observed in the abdominal cavity (Figure 3G).

Next, the morphology of the lesions was investigated by histological analysis to verify their endometrial character. The general morphology was assessed using hematoxylin and eosin (H&E) staining, which indicated the presence of blood vessels (Figure 4A). This was further confirmed by the positive staining for vimentin, illustrating the presence of both endothelial and stromal cells (Figure 4B, Appendix A). The presence of epithelial cells was confirmed by cytokeratin staining (Figure 4C, Appendix A). Lastly, growth associated protein 43 (GAP43) identified the presence of nerve fibers (Figure 4D, Appendix A) [13]. Non-adhered material was defined as vimentin and cytokeratin positive tissue without blood flow or innervation, as illustrated by the morphology (*i.e.,* H&E) and a negative GAP43 staining (Appendix A).

### 2.3. Ongoing Endometriosis-Associated Pain in Recipient Rats

After the injection of menstrual tissue to induce endometriosis, animals were followed up for 12 weeks using three different non-invasive pain assays, i.e., wellbeing, automatic dynamic weight bearing (ADWB) and open field test. Since earlier studies have reported the effects of estrogen levels on animal behavior [14], recipient animals were dosed with high E2 levels (500 ng/100 µL) prior to the behavior tests to synchronize the estrus cycle (Figure 3A). Two weeks after the endometriosis induction, weight gain of the ENDO recipient rat was significantly less compared to the SHAM group when normalized to baseline values (110.9 ± 0.7% vs. 115.1 ± 0.9% (mean ± SEM)). This difference was maintained until the end of the experiment (122.9 ± 1.1% vs. 130.1 ± 1.4% (mean ± SEM), ENDO vs. SHAM) (Figure 5A). Moreover, ADWB measurements indicated that after 12 weeks, the ENDO group spent significantly more time on all four paws (177.4 ± 12.5% vs. 134.8 ± 14.8% (mean ± SEM), ENDO vs. SHAM) (Figure 5B) and shifted its bodyweight more on the front paws compared to the baseline (169.6 ± 22.2% vs. 97.2 ± 12.6% (mean ± SEM), ENDO vs. SHAM) (Figure 5C). When assessing stress or anxiety of the animal using an open field test, no difference could be observed over time in total distance nor number of line crossings in the arena (Figure 5D,E). Note that all treated animals were included in the analysis, irrespective of the presence or absence of endometriosis lesions at the time of dissection (i.e., week 12).

### 2.4. Supplementation of Neuropeptides has no Effect on Endometriosis-Associated Pain

A recent study reported that deep endometriosis lesions could be reproduced in vivo by supplementing the lesions with the neuropeptides substance P (SP) and/or calcitonin gene-related peptide (CGRP) [15]. Therefore, osmotic pellets containing SP, CGRP or the combination of SP&CGRP were implanted in the nape of the neck of recipient animals, two weeks after ENDO or SHAM induction. Successful release of the substances and uptake in the circulation was evidenced by increased plasma levels of the neuropeptides two and six weeks after pellet implantation (Appendix A). When assessing the effect of neuropeptides on the presence of endometriosis lesions, both adhered and non-adhered tissue could be observed in the abdominal cavity after 12 weeks. In 55%, 50% and 30% of the animals in the SP, CGRP and SP&CGRP ENDO groups, respectively, adhered lesions were observed. However, the number of adhered lesions was not significantly higher in the group supplemented with neuropeptides compared to the regular treated ENDO group (Figure 6). Likewise, the size of adhered lesions did not differ significantly after neuropeptide supplementation. The histological analysis of these lesions revealed no particular changes in morphology via H&E staining or cellular composition (i.e., vimentin, cytokeratin and GAP43 positive) in comparison to the regular protocol (Appendix A). Only the endometriosis lesions retrieved from the SP group showed lower GAP43 positivity.

In addition, SP, CGRP and SP&CGRP supplemented animals were analyzed for ongoing pain. However, no differences were detected in the body weight gain between ENDO and SHAM animals in neuropeptide supplemented groups (Appendix A). Likewise, no differences were observed between the groups in postural changes or pain-related anxiety (Appendix A).

## 3. Discussion

The impact of ongoing or chronic pain on a patient’s physical and mental health cannot be underestimated, as it can impair daily tasks and lead to isolation from society [16]. Therefore, there is a need for animal models that can be used to examine the pain pathophysiology in closer detail, and could be used in drug discovery projects. In literature, numerous animal models for endometriosis are described; however, only a minority investigates the ongoing pain. Here, we aimed to introduce a rat model for endometriosis to investigate endometriosis-associated pain. The rat was selected as animal model, since most behavioral assays have been optimized for this species. The main hurdle with rodents, however, is the lack of spontaneous menstruation, which is assumed to be the key-event for endometriosis pathogenesis. Therefore, a menstruating rat model was developed based on earlier reported menstruating models in mice [9,11].

Decidualization comprises macroscopic (i.e., morphologic) and microscopic (i.e., biochemical and vascular) changes driven by the estrogen and progesterone receptors [17]. As the morphologic changes upon decidualization are evident, the uterine weight is regarded as a relevant marker to evaluate the decidualization process. Indeed, MRM was able to increase the uterine weight of a stimulated horn by four-fold compared to the control horn, which is in line with previous murine reports [12,18,19]. Likewise, the endometrial surface increased over 10-fold compared to the control. Therefore, there is also an alteration of the normal morphology of the endometrium, i.e., the lumen is reduced due to an expansion of the stromal compartment. In addition, the stromal compartment of the decidualized horn showed positivity for desmin, confirming the differentiation of the endometrial stromal cells into decidual cells [20]. The success rate of the MRM to induce decidualization in rats, i.e., 84.5%, is also comparable to other studies, as these describe signs of decidualization in 70% to 100% of the treated animals [18,19,21]. Additionally, 22.2% of the treated horns showed a partial or uneven decidualization process, which is again in consonance with literature and can be attributed to the dynamic nature of this reaction [19,21]. The biochemical and vascular changes upon decidualization were investigated using RT-qPCR. Here, genes that are involved in characteristic menstrual processes, such as the synthesis of prostaglandins, vasoactive and inflammatory mediators, extracellular matrix-modifying enzymes and leukocyte influx are upregulated [12,22]. More specifically, *Ptgs1* expression did not alter, whereas the inducible *Ptgs2* expression increased significantly in both the decidual and menstrual tissue. These genes encode the enzymes cylooxygenase-1 and -2 (Cox-1 and -2), respectively. The former is regarded to serve ‘housekeeping’ functions as a constitutive enzyme, whereas *Cox-2* is highly inducible by e.g., proinflammatory responses such as implantation [23]. Furthermore, an upregulation of the vasomodulative gene *Vegfa* was observed in the menstrual tissue compared to the control tissue. This is in line with the expectations, as decidual angiogenesis is mainly regulated through Vegfa secreted by decidual stromal cells [24]. During menstruation, matrix metalloproteinases are expressed, which either facilitate the tissue destruction in a first phase, or the reepithelization in the later part [12]. Indeed, only 4 h after progesterone withdrawal, an upregulation of *Mmp1*, *-3*, *-9* and *-10* was observed in the menstrual tissue. This upregulation is not as apparent in the decidual tissue, as progesterone withdrawal is a known regulator of the production of MMP production [25]. Lastly, the expression of *Tnf*, *Il6* and *Cxcr2* was increased, which are all known to peak in expression during the menstrual phase due to leukocyte infiltration [26]. Overall, these results confidently showed that the MRM generates representative menstrual endometrial tissue, which was then used to induce endometriosis in recipient animals.

Indeed, a single i.p. injection of menstrual endometrial tissue resulted in a successful induction of endometriosis in recipient animals, thereby re-enacting retrograde menstruation as seen in women. In 45% of the treated animals, adhered macroscopic lesions were observed in which neuro-angiogenesis was present, as illustrated by H&E morphology, and the presence of vimentin, cytokeratin and GAP43 positive cells. Although key-features of endometriosis were observed upon histological analysis, no fibromuscular stains were performed, as suggested by Vigano and colleagues, who defined endometriosis as “a fibrotic condition in which endometrial stroma and epithelium can be identified” [27]. However, the disease incidence was reduced in comparison to the endometriosis mouse model of Greaves et al. [9]. This may be attributed to the relatively low inoculation mass per body weight. This parameter was approximately 2 mg/g body weight in the mouse model [9], compared to 1.5 mg/g body weight in the rat endometriosis model. In accordance with Dodds et al., lesions were most often observed on the stomach and pancreas, although lesions could also be retrieved from other parts of the abdominal cavity (i.e., gonadal adipose tissue and intestines) [28]. The presence of adhered endometriosis lesions on the peritoneum or subcutaneously is most likely due to residual tissue fragments that were on the tip of the catheter when withdrawing the instrument from the abdominal cavity [28]. The proliferation of the endometriosis lesions was left under the control of the natural estrus cycle of the recipient animals. Retaining the natural cycle results in cyclic exposure of the endometriosis lesions to estrogen, as seen in endometriosis patients, which is known to be of the utmost importance in the thriving of lesions in the abdominal cavity, as proliferation occurs in an estrogen-dependent manner [29]. Furthermore, during the estrus cycle, *Vegf* is expressed in endometrial tissue and is involved in the estrogen-induced increase of permeability and proliferation of uterine blood vessels [30]. Therefore, when the endometriosis lesions are subjected to the estrus cycle, the expression of *Vegf* will generate neuro-angiogenic cues [31], supporting our goal to establish an endometriosis model for ongoing pain. This is in contrast to the mouse model, where recipients continuously received estrogen via a silastic implant, enabling proliferation of the lesions [9]. These implants, however, are also associated with urinary retention and bladder cystitis [32], two phenotypes that might interfere with the endometriosis-associated pain measurements. By relying on the natural estrus cycle, the use of estrogen pellets becomes redundant, thus evading these interfering bladder pain phenotypes.

Not all of the injected tissue, however, adheres to the surrounding organs. Some remains in the abdominal cavity, resulting in floating tissue [33]. Even though this is the case in 22.5% of the treated animals, we cannot exclude that these animals did not develop microscopic lesions. Likewise, animals with no adhered or non-adhered tissue, cannot be regarded as shams or healthy controls, as our examination for endometriosis lesions occurs by visual/macroscopic confirmation. Microscopic lesions or alterations might be missed, but these could still be innervated, thus causing endometriosis-associated pain. For this reason, all treated animals were included in the ENDO group, when investigating ongoing pain.

With regards to measuring chronic pain in laboratory animals, some considerations must be made. In many studies, acute assays (e.g., hotplate tests, Von Frey assay) serve to detect chronic pain, merely by repeating these experiments over time [34]. Therefore, the testing protocol must be chosen carefully to diminish conditioning of the animals. Additionally, it is important that the experimental design reflects a clinically relevant complaint [34]. In general, reflexive withdrawals to evoking stimuli (heat, cold or mechanical) are used for pain research to investigate mechanical and temperature hypersensitivity [34]. Although hypersensitivity is seen in chronic pain patients, this symptom is often less burdensome in comparison to ongoing pain. Therefore, in the present study, we assessed spontaneous ongoing pain using three independent pain assays, as categorized by Mogil et al. [34]: (i) a biomarker (i.e., body weight), (ii) functional disability (i.e., ADWB) and (iii) pain-depressed behavior (i.e., open field assay). An increase in weight is expected over the course of the experiment, as the animals gradually mature during the protocol. However, the first assay showed that the body weight gain of the ENDO group was significantly less as soon as week 2 after ENDO induction compared to SHAM animals. Secondly, ADWB measurements detected differences in the posture of ENDO compared to SHAM. Indeed, by week 12, rats from the ENDO group spent more time on four paws and appeared to shift their weight onto their front paws. This type of postural changes has already been described in relation to abdominal pain, as Laux-Biehlmann et al. showed that mice with zymosan-induced peritonitis increase their front/rear paw ratio and decrease rearing time due to abdominal discomfort [35]. Finally, an open field assay was performed to investigate alterations in mobility or explorative behavior. However, no differences regarding the locomotor activity was observed. Neither the total distance, nor the number of line crossings significantly decreased in the ENDO group compared to the SHAM. It has, however, been stated by Mogil et al. that open-field environments render it difficult to distinguish between explorative behavior, anxiety or disability [34]. Therefore, additional behavioral tests, such as elevate plus maze and/or novelty-suppressed feeding, should be included in the experimental design to investigate the anxiety/depression in more detail. Note that when only animals with macroscopic adhered endometriosis lesions were compared to sham controls, in a retrospective analysis, similar differences in pain behavior were observed: body weight gain and front/rear paw ratio differ significantly between both groups at week 12, while a trend to increase in the time on four paws was observed (*p* = 0.052).

In an effort to further optimize the MRM for endometriosis-associated ongoing pain, the effect of neuropeptide supplementation was investigated, as was described earlier [15]. Overall, no significant differences were observed in the number of adhered lesions, nor in their cellular composition. The gross of the lesions was vascularized, although not all displayed GAP43 positivity. Additionally, no differences were observed in the severity of the endometriosis-associated ongoing pain between ENDO and SHAM groups when supplemented with SP, CGRP or SP&CGRP. The fact that identified lesions were not hyper-innervated following neuropeptide supplementation could potentially explain the lack of increased pain symptoms.

In conclusion, we developed a rat model for endometriosis, in which we successfully induced endometriosis lesions derived from donor rat menstrual tissue. The recipient rats showed obvious signs of ongoing pain 12 weeks after induction, which are reminiscent of complaints of women with endometriosis. However, it must be stated that this model does not reflect the patient’s fertility problems, nor is it able to represent the asymptomatic populations. This novel research model may become an important tool to further investigate endometriosis-associated ongoing pain, instrumental in the development of therapies for treatment of endometriosis-associated ongoing pain. 

## 4. Materials and Methods

### 4.1. Animals

All animal experiments were approved by the local Ethical Committee for Animal Experimentation of the Faculty of Medicine of the KU Leuven (P101/2018, approved on 17 July 2018 and 22 July 2019). Rats (Sprague Dawley (Janvier, Le Genest-Saint-Isle, France)) were housed in ventilated cages and kept under controlled conditions. All animals were 8 to 10 weeks old at the start of the experiments.

### 4.2. Menstruating Rat Model 

Menstruation was induced in adult female donor rats using a protocol based on the menstruating mouse model of Cousins et al. [11] and Greaves et al. [9] (Figure 1A). In brief, rats underwent ovariectomy at day 0, under isoflurane-induced anesthesia (2% Iso-Vet; Eurovet, Bladel, the Netherlands). At the beginning of the surgery, a subcutaneous injection of 0.3 mg/kg buprenorphine (Vetergesic; Ecuphar, Breda, the Netherlands) was given as a painkiller. After one week of recovery, the uteri were primed with subcutaneously injections of high estradiol-17β (500 ng/100 µL peanut oil (Sigma-Aldrich, Diegem, Belgium)) on three consecutive days (i.e., day 7–9, ±9 a.m.). In order to mimic the window of implantation, a progesterone-releasing pellet (3 cm; fabricated in-house using SILASTIC Down Corning Corp, Midland, MI, USA; Sigma-Aldrich, Diegem, Belgium) was implanted in a subcutaneous pouch on day 13 (±14 p.m.). Note that prior to use, the pellets were incubated overnight at 37 °C in a 5% fetal bovine serum solution (Gibco, Merelbeke, Belgium). Furthermore, another series of low estradiol-17β injections (5 ng/100 µL peanut oil) was given subcutaneously at day 13–15 (±9 a.m.). The implantation/decidualization stimulus was given at day 15 (±14 p.m.) under isoflurane-induced anesthesia. Concisely, a small lateral midline skin incision was made caudal to the posterior border of the ribs. Another small lateral incision was made in the peritoneum, opening the abdominal cavity. Using a 20 G needle, the distal end of the uterine horn relatively to the cervix was punctured. Through this, a blunted 22 G spinal needle (modified in-house; Insyte, Vialon, Becton Dickinson, Madrid, Spain) entered the uterine lumen, giving a mechanical stimulus by scratching the endometrial surface three times and injecting 200 μL of sesame oil. Four days later (day 19 ± 9 a.m.), the pellet was removed to initiate progesterone withdrawal, which subsequently resulted in menstrual endometrial tissue 4 h later. Animals were sacrificed at different time points during the MRM, after which the endometrial tissue was collected. Contingent upon the analysis, tissue was treated differently: for histologic analysis, the complete uterine horn was fixed in 4% paraformaldehyde (PFA). For RNA extraction, endometrial tissue (without myometrium) was placed in RNALater (Qiagen, Venlo, the Netherlands) for 48 h and stored at −80 °C until further RT-qPCR analysis.

### 4.3. Vaginal Smear Examination

Methods were based on established protocols from our research group [36]. Briefly, the vaginal cytology was examined to determine the correct estrus phase of the animals. Using 100 µL phosphate buffered saline (PBS), the vagina was gently flushed two–three times, after which the sample was assembled on a glass slide for further microscopic examination. Under a bright field microscope using a 10× objective, the proportion among nucleated epithelial cells, cornified epithelial cells and leukocytes was observed in the sample. Depending on this assessment, the estrus phase could be determined [37].

### 4.4. RT–qPCR Experiments

RT–qPCR experiments were performed on RNA isolated from freshly isolated rat uterine horns (EST *n* = 3; non-stimulated *n* = 3; DECI *n* = 3; MENS *n* = 3). The tissue was left for 48 h on the RNALater (Qiagen, Venlo, The Netherlands) and stored at −80 °C until further RT-qPCR analysis. By use of a power homogenizer (Polytron, Montreal, QC, Canada), the tissue was homogenized and total RNA was extracted with TriPure Isolation Reagent (Roche, Mannheim, Germany). RNA concentrations were assessed using the Nanodrop method (Isogen Life Science, Temse, Belgium) and RNA quality was assessed using an Experion RNA StdSens Analysing kit (Bio-Rad, Nazareth Eke, Belgium) (good quality RNA samples included an RNA integrity number 7). Subsequently, 1 µg RNA was used for cDNA synthesis using the High-Capacity cDNA Reverse Transcription Kit (Life Technologies Europe B.V., Ghent, Belgium), and Triplicate cDNA (20× diluted) samples from each independent preparation were used in the StepOne PCR system (Applied Biosystems, Life Technology, Carlsbad, CA, USA) using Custom TaqMan Array Fast Standard Plate 16 (10 µL reaction) for the genes of interest (Appendix A). For each well, 1.080 µL cDNA was added to Mastermix (Life Technology, Merelbeke, Belgium), resulting in a final volume of 2 µL. *Actb*, *Gapdh* and *Tbp* were used as endogenous controls. The protocol consisted of a holding stage at 95 °C for 20 min followed by a cycling stage of 40 replication cycles at 60 °C for 20 min (StepOne PCR system, Applied Biosystems, Life Technology). Genes with Cq values above 35 cycles were considered as non-detectable. Data were shown as 2^(−ΔCq)^ (mean ± SEM) in which ΔCq = Cq_gene_ − Cq_geometric mean of endogenous controls_. Statistical tests were performed on the ΔCq values.

### 4.5. Endometriosis Induction

Endometriosis was induced in adult female recipient rats using a protocol based on the menstruating mouse model of Greaves et al. [9]. Briefly, menstrual endometrial tissue from the donor’s decidualized horns (cultivated via the MRM) was separated from the myometrium and finely minced using scissors (size of the pieces < 18 G). Any individual variability was minimized by pooling the tissue together in saline, before dividing the tissue into different syringes. In general, a single uterine horn was used to induce endometriosis in one recipient rat. Approximately 400 mg wet weight tissue/0.6 mL saline was injected i.p. in each recipient rat using a 16 G catheter. Note that the recipient rats were given high estradiol-17β (500 ng/100 µL peanut oil) two consecutive days beforehand to synchronize the estrus cycle (Figure 3A). Twelve weeks after the inoculation, recipient rats were sacrificed and the presence of endometriosis was assessed. Lesions were fixed in 4% PFA for histologic analysis. SHAM animals received an i.p. injection of 0.6 mL saline.

### 4.6. Immunohistochemistry

All immunohistochemistry pictures were taken on a Nikon Eclipse C*i* microscope.

#### 4.6.1. Hematoxylin & Eosin

Standard hematoxylin and eosin staining was used for the assessment of the histological and morphological features of the uteri and lesions. Briefly, 4 μm sections were subjected to a series of deparaffinization and rehydration, respectively toluene and 100% ethanol. Nuclei were stained with Gill’s haematoxylin (Prosan, Merelbeke, Belgium) during 4 min, followed by a few dips in acid alcohol (1% HCl in ethanol), and subsequently, lithium carbonate (saturated in Aqua distillate (AD)). Cytoplasmic staining was achieved with eosin (Prosan, Belgium) during 3 min. Note that sections were rinsed in tap water, and subsequently, AD between each of the previous steps. Sections were then dehydrated in graded alcohols, cleared in xylene and coverslips were mounted with a histological mounting media (Depex mounting medium, BDH Prolabo, The Netherlands).

#### 4.6.2. Cytokeratin, Vimentin and GAP43

In general, endogenous peroxidases were blocked with 3% H_2_O_2_ in methanol for 30 min followed by antigen retrieval. Blocking of non-specific binding was attained by an hour incubation with a block buffer containing 2% bovine serum albumin (Sigma-Aldrich, Belgium), 1% non-fat dry milk (Nestlé, Anderlecht, Belgium) and 0.1% Tween80, unless mentioned differently. Subsequently, the primary and secondary antibody were incubated and morphology was visualized by counterstaining with hematoxylin. Note that the slides were washed with Tris buffered saline between each step. Table 1 displays the specifics of each assay.

### 4.7. Behavioral Tests

To eliminate variability in the behavioral responses of the female rats due to differences in the estrus cycle, recipient rats were given high estradiol-17β (500 ng/100 µL peanut oil) two consecutive days beforehand to synchronize the estrus cycle (Figure 3A).

#### 4.7.1. Advanced Dynamic Weight Bearing Assessment

The ADWB device (Bioseb, Boulogne, France) was a Plexiglas enclosure in which an animal can move freely. The animal was placed into the ADWB device for a duration of 5 min, without a previous acclimatization period, to maximize exploratory behavior. A matrix comprising nearly 2000 high precision force sensors was embedded in the floor of the enclosure. The force sensors measured the weight distribution on each paw of the animal. Simultaneously, the animal was recorded from above using a high definition camera. The video feed and the pressure data were colligated using the ADWB software v1.4.2.98. This way, the location of each paw during the entire duration of the test can be automatically correlated with the correct activated force sensors. Table 2 displays the parameters used to validate a zone.

#### 4.7.2. Open Field Assay

The open-field consists of an arena (100 × 100 × 38 cm) monitored by an automated video tracking system ANY-maze v4.99 (Dublin, Ireland). The software divides the arena virtually in a grid of 25 squares. Rats were placed in the center of the arena and allowed to move freely in the enclosure for 5 min. For further analysis, the following parameters were automatically calculated using the ANY-maze software: distance moved and number of line crossings.

### 4.8. SP and CGRP Supplementation

#### 4.8.1. Neuropeptide Pellets and Plasma Collection

Alzet osmotic pumps (model 2006, Charles River, Belgium) were implanted subcutaneously in recipient rats, two weeks after the endometriosis induction to chronically deliver SP or CGRP (Sigma-Aldrich, Diegem, Belgium) for 6 weeks at a rate of 50 µg/kg body weight/day. Tail vein blood samples were collected at week 0, week 4 and week 8 in Multivette coated with lithium heparin (Sarstedt, Antwerpen, Belgium) and spun at 1500× *g* for 10 min to retrieve the plasma fraction. This was stored at −20 °C until further analysis.

#### 4.8.2. SP & CGRP Plasma Concentration

Plasma concentrations of SP and CGRP were determined using the colorimetric enzyme linked immunosorbent assay kits (Abcam, Cambridge, UK; Bertin Bioreagent, Montigny le Bretonneux, France) following the manufacturer’s instructions.

### 4.9. Data Analysis and Display

For data display and statistical analysis, Graphpad Prism 8.3 (Graphpad software incorporated, La Jolla, CA, USA) was used. Endometrial and desmin surface areas were calculated using ImageJ v1.52.

## Figures and Tables

**Figure 1 ijms-21-02326-f001:**
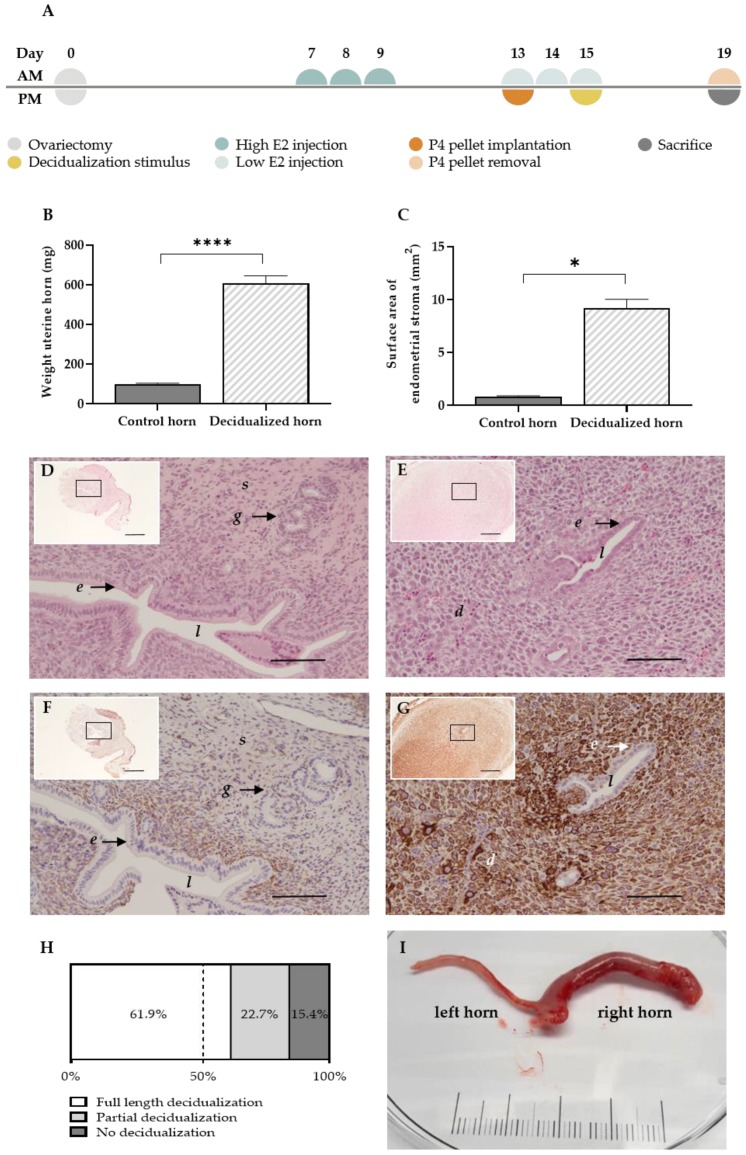
The MRM triggers decidualization successfully. (**A**) Experimental timeline of the MRM (**B**) Graph bar comparing the mean uterine weight (*n* = 18 rats) and (**C**) endometrial surface area at day 19 between decidualized and non-decidualized paired control horns (*n* = 7 rats). (**D**,**E**) H&E staining of a control and decidualized horn, respectively (scale 100 µm, scale of insert 500 µm). (**F**,**G**) Desmin staining in the control and decidualized horn, respectively (scale 100 µm, scale of insert 500 µm). (**H**) Full length, partial or no decidualization was observed in 61.9%, 22.7% and 15.4%, respectively, of all stimulated horns (*n* = total of 97 single horns, derived from 62 rats). (**I**) Representative example of decidualization at day 19 of the MRM: control left horn and decidualized right horn (scale in cm). *d* decidual cells; *e* luminal epithelial cells; *g* glandular epithelial cells; *l* lumen; *s* stromal cells. Data presented as mean ± SEM; Statistical differences in uterine weight were assessed using a paired *t*-test, ***** p* < 0.0001; Statistical differences in endometrial surface area were assessed using a Wilcoxon test, ** p* < 0.05.

**Figure 2 ijms-21-02326-f002:**
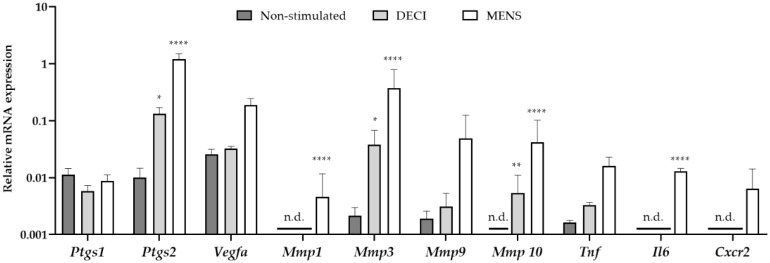
mRNA expression of menstrual genes in the endometrial tissue after the MRM. cDNA was synthesized from endometrial tissue at day 19 of the protocol, which was non-stimulated but undergoing MRM and keeping the P4 pellet (non-stimulated; *n* = 3), undergoing MRM and keeping the P4 pellet (DECI; *n* = 3), or undergoing MRM with P4 withdrawal (MENS; *n* = 3). Messenger RNA levels were quantified to the geometric mean of housekeeping genes *β-actine (Actb), Glyceraldehyde 3-phosphate dehydrogenase (Gapdh)* and *TATA-binding protein (TBP)*. *Mmp1, Mmp10, Il6* and *Cxcr2* were around (30 < Cq < 35) or below (Cq > 35) the detection limit, and were indicated as not determined (n.d.). Data are presented as mean ± SD. Statistically significant changes in mRNA expression are compared to the non-stimulated group and were assessed using a two-way ANOVA statistical test with Bonferroni correction, ** p* < 0.05, *** p* < 0.01, ***** p* < 0.0001.

**Figure 3 ijms-21-02326-f003:**
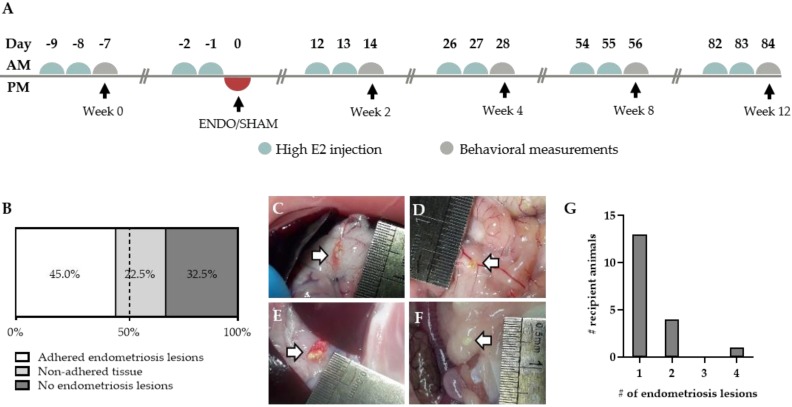
Endometriosis lesions in the abdominal cavity of recipient animals. (**A**) Experimental timeline of recipient animals. (**B**) Proportion of animals showing adhered, non-adhered and no endometriosis lesions (*n* = 40 rats). (**C**–**E**) Adhered endometriosis lesions (white arrow) can be found on the surrounding tissue, varying in size, color and location. (**F**) Non-adhered tissue is defined as white tissue that is floating in the peritoneal fluid. (**G**) The amount of adhered endometriosis lesions per animal (total number of animals *n* = 13).

**Figure 4 ijms-21-02326-f004:**
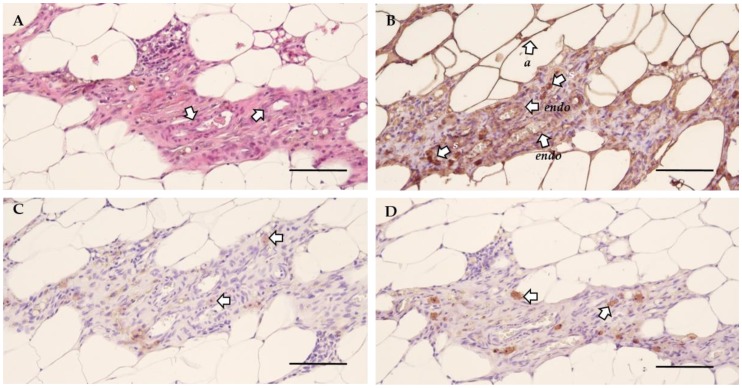
The cellular morphology of endometriosis lesions in rat. (**A**) H&E staining illustrating the general morphology and the presence of blood vessels (arrows) in the lesion. (**B**) Vimentin staining showed positivity for endothelial cells (*endo*), adipose tissue (*a*) and stromal cells (*s*). (**C**) Epithelial cells are identified by positivity for cytokeratin (arrows). (**D**) The presence of nerve fibers was confirmed by a positive GAP43 staining (arrows) (Scale 100 µm).

**Figure 5 ijms-21-02326-f005:**
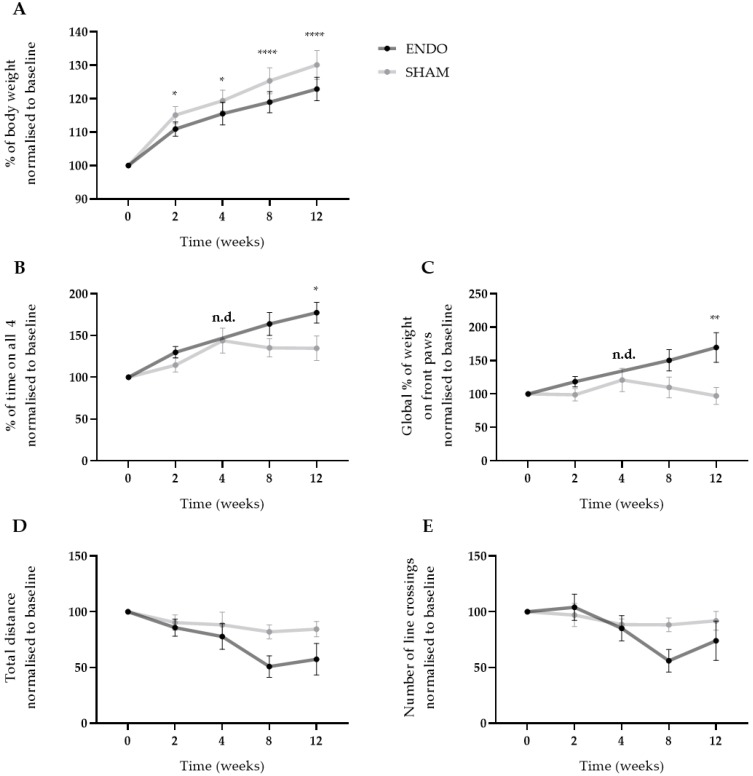
Ongoing endometriosis-associated pain can be observed in recipient rats. (**A**) The body weight gain of ENDO recipient rat decreased significantly over the course of 12 weeks compared to SHAM recipient rats. (**B**,**C**) Using ADWB, a significant increase in percentage of time on all four paws and global percentage of weight on the front paws was detected by week 12. Time point week 4 of the ENDO group was not determined due to technical problems with the sensor. (**D**,**E**) No differences were observed in the open field test when investigating the total distance and number of line crossings in the arena. ENDO *n* = 10, SHAM *n* = 9. Data presented as mean ± SEM. Statistical differences in the parameters of the behavior tests were assessed using a two-way ANOVA with Bonferroni correction, * *p* < 0.05, ** *p* < 0.01, **** *p* < 0.0001, n.d. not determined.

**Figure 6 ijms-21-02326-f006:**
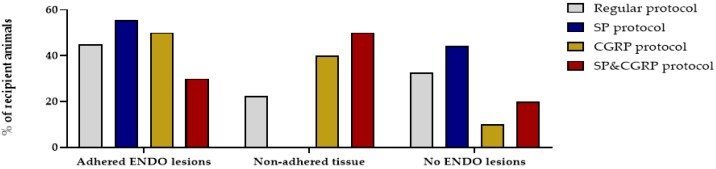
Supplementation of SP and/or CGRP does not affect the number or type of endometriosis lesions. Percentage of recipient animals with adhered ENDO lesions, non-adhered tissue or no ENDO lesions was not significantly different after the SP, CGRP or SP&CGRP protocol compared to the regular protocol (Regular *n* = 40; SP *n* = 9; CGRP *n* = 10; SP&CGRP *n* = 10 rats). Statistical differences were assessed using a two-way ANOVA with Bonferroni correction.

**Table 1 ijms-21-02326-t001:** Specifics of immunohistochemistry stains.

	Vimentin	Cytokeratin	GAP43
1^st^ Antibody	Anti-vimentin	Anti-rat cytokeratin 8	Anti-GAP43
Company-ref	Abcam-ab92547	Dako-Z0622	Abcam-ab75810
Marker	Endothelial and stromal cells	Epithelial cells	Nerve fibers
Type	Rabbit monoclonal	Rabbit polyclonal	Rabbit monoclonal
Blocking time	15′	15′	15′
Blocking	Regular	Regular + 1/30 NGS	Regular
Antigen retrieval	1 h at 90 °C in Tris EDTA	10′ at 37 °C in 0.04% pepsin in 0.01M HCl	1 h at 90 °C in citrate; pH = 6
Dilution 1^st^ Ab	1/1500	1/1500	1/250
Incubation 1^st^ Ab	1 h; 37 °C	2 h; RT	ON; 4 °C
2^nd^ Antibody	Goat anti-rabbit PO labelled	Goat anti-rabbit PO labelled	Swine anti rabbitHRP labelled
Dilution 2^nd^ Ab	1/100	1/100	1/400
Incubation 2^nd^ Ab	30′	30′	30′

NGS: normal goat serum; PO: peroxidase; HRP: horseradish peroxidase

**Table 2 ijms-21-02326-t002:** ADWB parameters

Zone Detection Parameters	
Low weight threshold (g)	1.00
Weight threshold (g)	2.00
Surface threshold	3
**Stable segment detection parameter**	
Minimum neighbor image	3
**Easy scoring**	
Stable images	15
Movement threshold	4

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
