# Peer review of "Mimicking Sampson’s Retrograde Menstrual Theory in Rats: A New Rat Model for Ongoing Endometriosis-Associated Pain"

_ijms, 2020, doi:10.3390/ijms21072326_

Round 1

Reviewer 1 Report

I commend the authors for taking on the challenge of designing a large rodent endometriosis model. It is certainly not easy!

I have a couple of queries about experimental design and a couple of suggested revisions.

Line 71; you highlight that the Greaves model has only be used to investigate central sensitization, have the authors try to use this model to look at ongoing pain symptoms? If not, what are the reasons not to use it? Or if the authors have looked at this, is this this reason they have decided to develop a rat model?

Line 83; Menstrual tissue generated in the Greaves endometriosis model utilises the menstruation protocol designed and optimised by Cousins et al., 2014 PLOS ONE. This was the first paper to use this hormone schedule and the 4 day decidualisation window. Please cite this paper accordingly here and on lines 253 and 396.

Line 93; change "scrapped" to "scraped"

Line 116; "of all stimulated horns (n=97 single horns)" does this equate to 97 rats and therefore 97 partial/fully decidualised horns and then 97 non-decidualised? Or have you counted both horns from some rats where the oil has induced decidualisation in both horns? Be clear in what you mean by "stimulated" versus decidualised. If you are just using these terms interchangeably then remove all references to stimulated. Or are you referring to all the horns you injected oil into irrespective of decidual status?

Line 125; What is your justification for using pro-oestrous endometrium which is under the influence of oestrogen and comparing it to DEC tissue which is under the influence of progesterone and also to MENS tissue which is undergoing progesterone withdrawal (and hasn’t seen exogenous oestrogen for 96 hours). Should these tissues have been compared to a metestrus tissue which is under progesterone regulation? Or better yet, why were the decidualised and menstruating horns not compared to their internal control? A horn which has been exposed to the exact same steroid hormones as the decidualised horn but has not undergone the decidual response?

Line 151; What is the justification for leaving lesions in for 12 weeks before dissection? Is this to model long term pain? Greaves et al look at 3 weeks, which I understand is "short term" but did the authors looks at 3, 6, 9 weeks to see how many lesions they get then? A 45% incidence rate at 12 weeks is not exactly a good model for long term disease, unless you already know if 100% developed endo and that it has cleared in 55% of rats or they didn't develop disease at all? I am very interested to know what the "take" rate is for lesions an earlier time point. Because how do you assess the 55% that didn't have lesions? Are those rats incorporated into the data displayed in Figure 5? Or is that just the 45% that did get lesions? If you compare the 45% versus the 55% do they exhibit similar pain scores irrespective of endo status?

Line 213; How many of your supplemented mice developed endo? Please include % in text.

Line 332; Answers my question line 151 i.e. 45% versus 55%, please include a line earlier in the manuscript to say that all rats were included in the pain analysis. Though it would be of interest to know if there were any differences if you separated them.

Line 549; Adhered lesions n=25, from how many recipients was this?

Line 566; Why was plasma concentration measured at 4 and 8 weeks and not at 12 weeks to know what the concentration was at time of dissection?

Author Response

I commend the authors for taking on the challenge of designing a large rodent endometriosis model. It is certainly not easy! I have a couple of queries about experimental design and a couple of suggested revisions.

We thank the reviewers for this supportive comment and have adjusted the manuscript accordingly.

Comments:

  • Line 71; you highlight that the Greaves model has only be used to investigate central sensitization, have the authors try to use this model to look at ongoing pain symptoms? If not, what are the reasons not to use it? Or if the authors have looked at this, is this reason they have decided to develop a rat model?

We thank the reviewer for this comment. The main focus of our research group is to investigate pathophysiology of pain. Investigating ongoing pain in animal models is somehow difficult, as not a lot of tests are available and rodents are prey species with strong motivation to hide their pain from potential predators (Mogil, The Oxford Handbook of the Neurobiology of Pain, The Measurement of Pain in the Laboratory Rodent, 2019). In addition, in line with the human situation, we expect endometriosis-associated pain to be subtle and vague. As such, identifying and quantifying these kind of pain symptoms in mice would pose additional complexity. Therefore, the use of a rat model was made out of practical consideration, as stated in the introduction (see line 77-81). In this study, we rely on the observation that abdominal pain or discomfort, as seen in endometriosis, would be translated in rodents to postural differences. Therefore, the automated dynamic weight bearing test was used. Additionally, the performance of this test has proven more effective in larger rodents, as the sensors can easier detect weight bearing differences in heavier subjects. For this reason, it was opted to develop a new rat model rather than testing the existing mouse models. We did not include the more classical behavior tests for mechanical and heat-hyperalgesia, as it is believed that these effects are caused by adaptations in the central nervous system. The main scope was to develop an animal model mimicking the human pain complaints.

  • Line 83; Menstrual tissue generated in the Greaves endometriosis model utilises the menstruation protocol designed and optimised by Cousins et al., 2014 PLOS ONE. This was the first paper to use this hormone schedule and the 4 day decidualisation window. Please cite this paper accordingly here and on lines 253 and 396.

We agree with the reviewer and paper references were inserted appropriately. 

  • Line 93; change "scrapped" to "scraped"

Errors were adapted accordingly.

  • Line 116; "of all stimulated horns (n=97 single horns)" does this equate to 97 rats and therefore 97 partial/fully decidualised horns and then 97 non-decidualised? Or have you counted both horns from some rats where the oil has induced decidualisation in both horns? Be clear in what you mean by "stimulated" versus decidualised. If you are just using these terms interchangeably then remove all references to stimulated. Or are you referring to all the horns you injected oil into irrespective of decidual status?

We agree with this comment and have clarified accordingly (line 126). Following definitions were used: stimulated means that oil was injected in the lumen of the uterine horn, and is irrespective of the decidual status. Stimulation of the uterine horn at day 15 can result in either full, partial or no decidualization of the horn at day 19. Therefore, the terms stimulated and decidualized are not interchangeable. This was further clarified in the text, see lines 104-106: Note that stimulation of the uterine horn (i.e. oil was injected in the lumen of the uterine horn at day 15) can have varying results in terms of the decidualization grade at the end of the protocol.’

  • Line 125; What is your justification for using pro-oestrous endometrium which is under the influence of oestrogen and comparing it to DEC tissue which is under the influence of progesterone and also to MENS tissue which is undergoing progesterone withdrawal (and hasn’t seen exogenous oestrogen for 96 hours). Should these tissues have been compared to a metestrus tissue which is under progesterone regulation? Or better yet, why were the decidualised and menstruating horns not compared to their internal control? A horn which has been exposed to the exact same steroid hormones as the decidualised horn but has not undergone the decidual response?

We appreciate the reviewer for this valid comment. In this experiment, we were mainly interested in gene expression difference caused by ’menstruation’, as menstrual tissue will appear in the peritoneal cavity in human. Therefore, we opted to compare MENS tissue with DECI, as the only difference here is the progesterone withdrawal that is required for the induction of menstruation. As we already observed the expression of some markers for ‘menstruation’ in the DECI tissue, we wanted to investigate whether these markers were caused by the decidualisation stimulus or whether they are constituently expressed in the endometrium. Although not presented in the previous manuscript, the authors performed analysis on DECI tissue compared to its internal control, i.e. tissues subjected to the MRM but without receiving the decidualization stimulus and kept the progesterone pellet. The original data containing the pro-estrus data was replaced with the internal control data (see Figure 2 of the revised manuscript) and the text was adapted accordingly, see lines 133-143 and 145-147. Interestingly, some markers showed significant differences. Therefore, we further investigated whether expression of the latter was due to the MRM, by comparing the non-decidualized horn to naïve endometrium collected during a normal estrous cycle. Because we wanted to know whether these genes were expressed because of the increased progesterone concentrations, we choose to compare it to naïve tissue that was not subject to progesterone. Therefore, we isolated endometrium from the pro-estrous phase. These data were also added as supplementary figure (figure S1) and the manuscript as adapted accordingly, see lines 155-164 and 574-583.

Line 151; What is the justification for leaving lesions in for 12 weeks before dissection? Is this to model long term pain? Greaves et al look at 3 weeks, which I understand is "short term" but did the authors looks at 3, 6, 9 weeks to see how many lesions they get then? A 45% incidence rate at 12 weeks is not exactly a good model for long term disease, unless you already know if 100% developed endo and that it has cleared in 55% of rats or they didn't develop disease at all? I am very interested to know what the "take" rate is for lesions an earlier time point. Because how do you assess the 55% that didn't have lesions? Are those rats incorporated into the data displayed in Figure 5? Or is that just the 45% that did get lesions? If you compare the 45% versus the 55% do they exhibit similar pain scores irrespective of endo status?

The lesions were left to grow in vivo over a period of 12 weeks to ensure that the extent of neuro-angiogenesis is sufficient, as the scope of this study was to investigate ‘ongoing pain symptoms over time’. Although the establishment of endometriosis lesions is an interesting topic, the longitudinal behavior experiments were prioritized and consequently no intermediate dissections were done. As stated in the discussion (line 354-362), determining if an animal has developed endometriosis was done by macroscopic evaluation. However, microscopic lesions and/or neuro-angiogenesis could have occurred as well, and could potentially contribute to the pain behavior. Therefore, all treated animals were included in the analysis. Furthermore, a retrospective analysis showed that when only the animals with macroscopic lesions are included in the analysis, similar differences in pain behavior were observed. As suggested by the reviewer, an additional analysis was performed wherein the ENDO group was divided in different subgroups (adhered ENDO lesions, non-adhered tissue and no ENDO lesions) and compared to the SHAM group. Here, however, no conclusion can be made as the sample sizes are too small for powerful statistical analysis (n = 5, 3 and 2, respectively).

  • Line 213; How many of your supplemented mice developed endo? Please include % in text.

We have included the percentage in text, see line 246-248.

  • Line 332; Answers my question line 151 i.e. 45% versus 55%, please include a line earlier in the manuscript to say that all rats were included in the pain analysis. Though it would be of interest to know if there were any differences if you separated them.

We hope to have clarified this comment, see line 224-226: Note that all treated animals were included in the analysis, irrespectively of the presence or absence of endometriosis lesions at the time of dissection (i.e. week 12).

  • Line 549; Adhered lesions n=25, from how many recipients was this?

This comment was clarified, the number of recipients was 18 (see line 590).

  • Line 566; Why was plasma concentration measured at 4 and 8 weeks and not at 12 weeks to know what the concentration was at time of dissection?

The main reason for not determining the neuropeptide concentration is of the limited lifetime of the osmotic pumps of 6 weeks. As it was deemed that the effect of the neuropeptides would be the highest during the initial phases of lesion adhesion and proliferation, and literature suggested to implant them during the early phases of the protocol (Yan, Hum Reprod, 2019), it was opted to implant the osmotic pumps 2 weeks after endometriosis induction. Plasma collection was performed at week 4 to verify that the plasma concentration was indeed elevated, and sustained until week 8. As the osmotic pump would not be functioning at week 12 of the protocol, baseline plasma concentrations were expected and therefore, no plasma was collected at this time point.

Reviewer 2 Report

  1. This interesting and useful manuscript reports the use of generated “menstrual” endometrial tissue for intraperitoneal injection in order to produce endometriosis lesions associated with pain, epithelial cells, stromal cells, neuro-angiogenesis and upregulation of Ptgs2, vascular endothelial growth factor a (Vegfa), Mmp 1, 3 and 10, 132 tumor necrosis factor (Tnf), interleukin 6 (Il6) and interleukin 8 recepor β (Cxcr2). The protocol included estradiol-17β days -2 &-1, ovariectomy day 0, estradiol-17β days 7-9, implanted progesterone-releasing pellet days 13-19, mechanical stimulus day 15, removal of progesterone-releasing pellet day 19, and harvesting 4 hours after removal on day 19. There is also an analysis of supplementing the lesions with neuropeptides that showed no effect on pain.
  2. Changes in bodyweight and posture (shifting weight to front paws) were seen as indicating the presence of ongoing, spontaneous pain. Consider discussing the additional historical observations of abdominally directed grooming (licking), exploratory activity, mechanical withdrawal thresholds, or mechanical allodynia as discussed in your reference #9 Greaves et al. Sci Rep. 621 2017;7:44169?
  3. Is pain on lines 28 and 344 “spontaneous” or “induced”? If “induced,” please change that and the associated discussion.
  4. “Results” beginning on line 81 includes results, methods, and discussion. Please move methods and discussion into the appropriate sections.
  5. Line 37. Is endometriosis “functional endometrium-like” or “dysfunctional endometrium-like.” Does endometriosis function in a normal fashion. The function of the endometrium is pregnancy. Can endometriosis support pregnancy?
  6. Figure 5A appears to show a weight increase in both groups rather than decrease. Are the legends correct? If so, please add an additional explanation.
  7. Figure 5B & 5C has “not determined” for the first four time periods. Do you mean “not significant,” that you did not test those, or something else? If so, please clarify that. If you did not test these, please test them? Please add a similar analysis for D & E.
  8. In humans, estrogens may increase the chance of implantation (Keettel & Stein Am J Obstet Gynecol. 1951, 61(2):440-442; Liang et al. Reprod BioMed Online. 2019, 38 (4):549-559, https://www.sciencedirect.com/science/article/pii/S1472648318306540. The progestin withdrawal of this study's protocol study does not include E2 stimulation on day 19 as occurs in the menstrual phase in a human. Why? Have you compared the characteristics of the endometriotic tissue in a MENS group to a MENS + estrogen group? If not, please specify that estrogen was not used in your protocol and why? How does this relate to your references #27 (Cullinan-Bove 1993), #28 (Sondell 1999), and # 26 Noble et al (1996)?
  9. Do you have endogenous E2 levels for the rats on day 19?
  10. Have you considered analyzing for a decrease or increase in endometriosis over time? A decrease is suggested by your references #3 Koninckx (Hum Reprod. 607 1994;9(12):2202-2205) and #4 Nap (Best Pract Res Clin Obstet Gynaecol. 2004;18(2):233-244). A decrease in superficial endometriosis and an increase in deep or other advanced forms may be responsible for the biphasic distribution in Koninckx’ Figure 1 (Fertil Steril 2019, 111:327–40, DOI: https://doi.org/10.1016/j.fertnstert.2018.10.013). However, those contrast with the progression discussed in D’Hooghe et al (Reprod Sci 16(2): 152-151 2009, DOI: 10.1177/1933719108322430) and his 1996 article (D’Hooghe et al Obstet Gynecol. 1996;88:462-466).
  11. Do you plan to look for a finding similar to the D’Hooghe et al (Reprod Sci 16(2): 152-151 2009, DOI: 10.1177/1933719108322430) induction of endometriosis in baboons can induce biological changes in their eutopic endometrium after 6 months?
  12. Your ref # 13 Yan, Liu, & Guo (Hum Reprod 34(2):235–247, 2019) concluded that there were no rodent models for deep endometriosis. Do you anticipate that your model might be used for deep endometriosis?
  13. Consider clarifying that a pain population may not represent infertility or asymptomatic populations.
  14. Sampson (Am J Obstet (and diseases of Women and Children) 1918, 78:161-75 and Am J Obstet Gynecol. 1927, 14(4):422-469) described both typical endometrium and “misplaced atypical endometrial tissue both in structure and in function.” Did you distinguish transplants of normal endometrium from those of abnormal endometrial-like (endometriotic) tissue? If so, how did you make that distinction?
  15. Koninckx’ endometriosis disease concept (Fertil Steril 2019, 111:327–40, Gynecol Obstet Invest 1999, 47(Suppl 1):3–9) appears to be supported in your reference #4 (Nap et al. Res Clin Obstet Gynaecol. 2004;18(2):233-244). In support of his concept, Koninckx questions whether “microscopic endometriosis and subtle lesions cause pain or infertility” (Fertil Steril 2019, 111:327–40). However your study, Koninckx’ biphasic pain curve in Figure 1 of that 2019 publication, and a body of pediatric publications since as early as 1980 (Goldstein t al. J Reprod Med 24(6):251-6, PMID: https://www.ncbi.nlm.nih.gov/pubmed/6448296) associate pain with early endometriosis. Do you feel your data support early endometriosis as a disease or do you feel that additional reactive, immunologic, biochemical, genetic, and epigenetic changes are needed before endometriosis is a disease?
  16. Do you feel your model is useful for both superficial and deep disease?
  17. Fibro-muscular stains are not listed. Did you use other markers for fibrosis or muscular metaplasia? Do you believe endometriosis needs fibrosis as a component as suggested by Vigano 2018 et al. (Hum Reprod. 2018, 33(3):347–352)?
  18. D’Hooghe et al. (Reprod Sci, 2009, 16:152-161) discussed that aromatase expression was only found after 10 months in endometriosis lesions in baboons (Fazleabas et al. Fertil Steril. 2003;80:820-827). Do you believe aromatase is required for a definition of endometriosis? See your Ref # 26 Noble et al (1996).
  19. The pages in reference #2 “Sampson JA. Am J Pathol. 1927;3(2):93-110.143” should be “93-110.43.”
  20. The use of your reference #2 “Sampson JA. Am J Pathol. 1927;3(2):93-110.143” is incorrect for “backward menstrual flow.” Sampson’s 1927 Am J Pathol reference is to vascular dissemination not to “backward menstrual flow.” You appear to need his 1927 AJOG reference “Sampson, John A. (1927) Peritoneal endometriosis due to the menstrual dissemination of endometrial tissue into the peritoneal cavity. Am J Obstet Gynecol. 1927, 14(4):422-469, https://doi.org/10.1016/S0002-9378(15)30003-X.
  21. Highlighting Sampson in the title invites examination of his several theories. I agree with you that Sampson’s retrograde theory is likely the major cause of endometriosis. In addition, in the 1927 AJOG article, he suggests the use of both retrograde dissemination and differentiation of celomic epithelium to explain the diffuse locations. In 1927 (AJP and AJOG) he discussed vascular dissemination, retrograde flow, and celomic metaplasia. In 1921, he discussed dissemination from hemorrhagic (chocolate) cysts (Arch Surg (now JAMA Surgery). 1921, 3: 245-323, https://jamanetwork.com/journals/jamasurgery/fullarticle/536143), In 1925 (Am J Obstet Gynecol 1925, 10:649-664), he reviewed and supported remnants from Wolffian bodies transplantation endometriosis and extraperitoneal metastatic endometriosis. In 1925 he also discussed the possibility of developmentally misplaced endometrial tissue, but stated he had “never been able to appreciate it.” 1925 may be the first publication of the term “endometriosis.” Since “Sampson’s Theory” is not adequately defined, this can be confusing and since your title is theory dependent, consider discussing his use of multiple theories. Consider an explanation like “Sampson postulated several theories of endometriosis development. Retrograde menstruation is the most likely to explain the location of the majority of cases of intraabdominal endometriosis (Sampson 1927a (AJOG), Sampson 1927b (AJP)).” Also consider modifying the title to “Mimicking Sampson’s Retrograde Menstrual Theory in Rats: A New Rat Model for Ongoing Endometriosis-Associated Pain”
  22. Sampson (AJOG 1927 stated that “both typical and atypical endometrial tissue was found and one could trace the transition of one type of lesion into the other.” Is your analysis of neuro-angiogenesis adequate to distinguish normal appearing transplanted endometrium from endometriosis as suggested by Koninckx’ endometriosis disease theory (Figure 2, Fertil Steril 2019, 111:327–40, DOI: https://doi.org/10.1016/j.fertnstert.2018.10.013)?
  23. Regarding “the oldest” hypothesis on line 39, the theories of 1927 were neither Sampson’s nor the literature’s oldest. Sampson had theories as least as early as 1921. An older review is in Russell (1899) who considered remnants of the germinal epithelium (Waldeyer, 1870), an extension of tubal epithelium (Marchand, 1879), a Wolffian body, and a Müller’s duct remnant. Russell WW. Aberrant portions of the Müllerian duct found in an ovary: Johns Hopkins Hospital Bulletin, 1899, 10:8-10 plus plates. If you search for that, you may link to Longo (Am J Obstet Gynecol 1979, 134(2):225-226) who reviewed Russell and other articles.

Author Response

Reviewer #2:

This interesting and useful manuscript reports the use of generated “menstrual” endometrial tissue for intraperitoneal injection in order to produce endometriosis lesions associated with pain, epithelial cells, stromal cells, neuro-angiogenesis and upregulation of Ptgs2, vascular endothelial growth factor a (Vegfa), Mmp 1, 3 and 10, 132 tumor necrosis factor (Tnf), interleukin 6 (Il6) and interleukin 8 recepor β (Cxcr2). The protocol included estradiol-17β days -2 &-1, ovariectomy day 0, estradiol-17β days 7-9, implanted progesterone-releasing pellet days 13-19, mechanical stimulus day 15, removal of progesterone-releasing pellet day 19, and harvesting 4 hours after removal on day 19. There is also an analysis of supplementing the lesions with neuropeptides that showed no effect on pain.

In general, we thank the reviewer for the detailed assessment of the manuscript and the constructive comments. Clearly, the reviewer is an expert in the field and addressing these comments will significantly improve the manuscript.

Comments:

  • Changes in bodyweight and posture (shifting weight to front paws) were seen as indicating the presence of ongoing, spontaneous pain. Consider discussing the additional historical observations of abdominally directed grooming (licking), exploratory activity, mechanical withdrawal thresholds, or mechanical allodynia as discussed in your reference #9 Greaves et al. Sci Rep. 621 2017;7:44169?

We would like to thank the reviewer for this comment but want like to highlight that this research focusses on ongoing, chronic pain symptoms. The suggested historical observations of exploratory activity were indeed performed (Figure 5D), although did not show a significant decrease in activity in the ENDO group. Mechanical and thermal allodynia, however, are rather a consequence of central sensitization, as stated in line 368-375, and thus was not investigated as this is outside the scope of the paper.

  • Is pain on lines 28 and 344 “spontaneous” or “induced”? If “induced,” please change that and the associated discussion.

As endometriosis does not occur spontaneously in rodents, spontaneous ongoing endometriosis-associated pain cannot be detected. Only after endometriosis induction, ongoing endometriosis-associated pain can be investigated. We have clarified these sections accordingly, see lines 29 and 372.

  • “Results” beginning on line 81 includes results, methods, and discussion. Please move methods and discussion into the appropriate sections.

We would like to clarify our rationale of layout. Many of the experiments of this research paper asks for more framework (cfr reviewer n° 1 who demanded for more elaborate explanations about the study design). Furthermore, this paper stipulates a new protocol to induce menstrual tissue, therefore, we choose to keep a short overview of the methods in the results section as a clarification for the reader.

  • Line 37. Is endometriosis “functional endometrium-like” or “dysfunctional endometrium-like.” Does endometriosis function in a normal fashion. The function of the endometrium is pregnancy. Can endometriosis support pregnancy?

By stating that endometriosis is functional endometrium-like, it is meant that lesions still show responsiveness to the menstrual hormones. Of course, the tissue is not able to sustain pregnancy. Therefore, we have altered these sections accordingly, see lines 16 and 38.

  • Figure 5A appears to show a weight increase in both groups rather than decrease. Are the legends correct? If so, please add an additional explanation. Figure 5B & 5C has “not determined” for the first four time periods. Do you mean “not significant,” that you did not test those, or something else? If so, please clarify that. If you did not test these, please test them? Please add a similar analysis for D & E.

An increase in weight is expected over the course of the experiment, as the animals are only 8-10 weeks old at the start of the protocol. In the ENDO group, however, the weight gain is significantly reduced compared to the SHAM group. In panel B and C, at week 4, the ADWB measurements of the ENDO group were not determined, due to technical problems of the ADWB sensor (hence the lack of the corresponding data point). In panel D and E, a similar analysis was performed as for the other panels, although no significant differences were observed.

We have clarified these sections accordingly, line 375-377: An increase in weight is expected over the course of the experiment, as the animals gradually mature. And line 234-235: Statistical differences in the parameters of the behaviour tests were assessed using a Two-way ANOVA with Bonferroni correction.

  • In humans, estrogens may increase the chance of implantation (Keettel & Stein Am J Obstet Gynecol. 1951, 61(2):440-442; Liang et al. Reprod BioMed Online. 2019, 38 (4):549-559, https://www.sciencedirect.com/science/article/pii/S1472648318306540. The progestin withdrawal of this study's protocol study does not include E2 stimulation on day 19 as occurs in the menstrual phase in a human. Why? Have you compared the characteristics of the endometriotic tissue in a MENS group to a MENS + estrogen group? If not, please specify that estrogen was not used in your protocol and why? How does this relate to your references #27 (Cullinan-Bove 1993), #28 (Sondell 1999), and # 26 Noble et al (1996)?

To the best of our knowledge, we have simulated the human situation as best as practically feasible in rodents. The days in the study protocol do not refer to the exact same day as the menstrual cycle. As such, day 7-12 mimic the proliferative phase (high estrogen peak), days 13-18 the secretory phase (high progesterone peak, low estrogen peak) of the menstrual cycle and day 19 after progesterone withdrawal mimics the menstrual phase (low estrogen, low progesterone). As this ‘human’ protocol is imposed on rodents, we try to combine essential steps from both species: hormonal regimes are mimicked but the timing is adjusted to the rodent situation. As such, we supplemented a low dose of estrogen on days 13-15 of the protocol together with progesterone administration, where after the decidualization stimulus was given to mimic the implanting embryo, which is in rodents essential to trigger decidualization. Successful decidualization observed in the protocol therefore means that the hormone regime was sufficient. We hope this clarifies the comment of the reviewer.

  • Do you have endogenous E2 levels for the rats on day 19?

We did not investigate the endogenous E2 levels in rat. As the animals had an ovariectomy and only received exogenous E2, the endogenous E2 levels are expected to be minimal by day 19, as this is the end of the cycle.

  • Have you considered analyzing for a decrease or increase in endometriosis over time? A decrease is suggested by your references #3 Koninckx (Hum Reprod. 607 1994;9(12):2202-2205) and #4 Nap (Best Pract Res Clin Obstet Gynaecol. 2004;18(2):233-244). A decrease in superficial endometriosis and an increase in deep or other advanced forms may be responsible for the biphasic distribution in Koninckx’ Figure 1 (Fertil Steril 2019, 111:327–40, DOI: https://doi.org/10.1016/j.fertnstert.2018.10.013). However, those contrast with the progression discussed in D’Hooghe et al (Reprod Sci 16(2): 152-151 2009, DOI: 10.1177/1933719108322430) and his 1996 article (D’Hooghe et al Obstet Gynecol. 1996;88:462-466).

The suggestions to investigate the biphasic distribution of endometriosis over time is interesting. However, the lesions that are developed in the rat model were not classified as superficial or deep endometriosis, as these were not (histologically) investigated for their respective markers, nor was the infiltration depth of the lesion measured.  Furthermore, the longitudinal behavior experiments were prioritized and no intermediate dissections were done. We thank the reviewer for this suggestion that could be implemented in the next follow up study.

  • Do you plan to look for a finding similar to the D’Hooghe et al (Reprod Sci 16(2): 152-151 2009, DOI: 10.1177/1933719108322430) induction of endometriosis in baboons can induce biological changes in their eutopic endometrium after 6 months?

We thank the reviewer for this interesting suggestion as it would be commended to investigate this in other models. However, the limitation of rodent models lies within the fact that implanted lesions are not able to undergo menstruation as seen in the human situation. Since decidualization, and thus menstruation, will only occur after application of a decidualization stimulus, such as an implantation embryo or the injection of oil, this cannot be performed on the endometriotic lesions within the abdomen. Most likely, factors secreted by this cycling endometrium might contribute to inducing changes in the eutopic endometrium, and this cannot be mimicked in rodent endometriotic lesions. Therefore, we believe this model holds great potential for investigating pain, whereas it might be limited to investigate other aspects of the pathophysiology of endometriosis.

  • Your ref # 13 Yan, Liu, & Guo (Hum Reprod 34(2):235–247, 2019) concluded that there were no rodent models for deep endometriosis. Do you anticipate that your model might be used for deep endometriosis?

There is indeed a need for research models that are able to selectively induce the different types of endometriosis. In our current model, however, we did not focus on lesion development, as mentioned in the previous comments, and consequently did not investigate this in further detail (i.e. extensive histological analysis of the lesions (e.g. fibro-muscular stains)). Furthermore, to date, there is no correlation described between the type of lesion or its location and the extend of the pain symptoms. Therefore, we deemed that if an animal model is able to have any type of lesions which are innervated, this is sufficient to investigate ongoing endometriosis-associated pain.

  • Consider clarifying that a pain population may not represent infertility or asymptomatic populations.

We agree with the reviewer and have implemented this comment in the discussion, see text line 412-414: However, it must be stated that this model does not reflect the patient’s fertility problems nor is it able to represent the asymptomatic populations.

  • Sampson (Am J Obstet (and diseases of Women and Children) 1918, 78:161-75 and Am J Obstet Gynecol. 1927, 14(4):422-469) described both typical endometrium and “misplaced atypical endometrial tissue both in structure and in function.” Did you distinguish transplants of normal endometrium from those of abnormal endometrial-like (endometriotic) tissue? If so, how did you make that distinction?

In this study, no distinction, per se, was made in the transplants to see if they can be categorized as normal endometrium or abnormal endometrial-like (endometriotic). However, in retrospect, we did observe that the epithelial cells were often not delineating a lumen and scattered in the lesion. Therefore, we would classify these lesions more as abnormal endometrial-like tissue (Donnez, Hum Reprod, 1996; Muzii, Fertil Steril, 2007; Vigano, Hum Reprod, 2018).

  • Koninckx’ endometriosis disease concept (Fertil Steril 2019, 111:327–40, Gynecol Obstet Invest 1999, 47(Suppl 1):3–9) appears to be supported in your reference #4 (Nap et al. Res Clin Obstet Gynaecol. 2004;18(2):233-244). In support of his concept, Koninckx questions whether “microscopic endometriosis and subtle lesions cause pain or infertility” (Fertil Steril 2019, 111:327–40). However your study, Koninckx’ biphasic pain curve in Figure 1 of that 2019 publication, and a body of pediatric publications since as early as 1980 (Goldstein t al. J Reprod Med 24(6):251-6, PMID: https://www.ncbi.nlm.nih.gov/pubmed/6448296) associate pain with early endometriosis. Do you feel your data support early endometriosis as a disease or do you feel that additional reactive, immunologic, biochemical, genetic, and epigenetic changes are needed before endometriosis is a disease?

We thank the reviewer for this interesting comment. Although it is most likely that additional reactive, immunologic, biochemical, genetic, and epigenetic changes are needed for the disease to further evolve and progress, we cannot exclude that patients experience pain symptoms in the early stages of endometriosis. Indeed, it is reported that patients with microscopic endometriosis and subtle lesions experience mild to extreme pain symptoms and infertility. However, additional research is required to clarify whether early endometriosis can be stated as a disease.  

  • Do you feel your model is useful for both superficial and deep disease?

As stated in the previous comments, no histological stains were performed to validate the type of endometriosis lesion in the rat model. Therefore, no conclusive remark can be made regarding the usefulness of the model to be used outside its potential to investigate endometriosis-associated pain.

  • Fibro-muscular stains are not listed. Did you use other markers for fibrosis or muscular metaplasia? Do you believe endometriosis needs fibrosis as a component as suggested by Vigano 2018 et al. (Hum Reprod. 2018, 33(3):347–352)?

As mentioned in the previous comments, fibrosis or muscular metaplasia was not investigated in light of this study. However, we have implemented this comment in the discussion, see text line 324-327: Although key-features of endometriosis were observed upon histological analysis, no fibromuscular stains were performed, as suggested by Vigano and colleagues, whom define endometriosis as “a fibrotic condition in which endometrial stroma and epithelium can be identified”.

  • D’Hooghe et al. (Reprod Sci, 2009, 16:152-161) discussed that aromatase expression was only found after 10 months in endometriosis lesions in baboons (Fazleabas et al. Fertil Steril. 2003;80:820-827). Do you believe aromatase is required for a definition of endometriosis? See your Ref # 26 Noble et al (1996).

We thank the reviewer for this comment but we cannot make any conclusive statements as we haven’t investigated the aromatase expression in the lesions. Literature indeed suggests that the aromatase expression is different in the ectopic lesions compared to the eutopic endometrium. However, it is not clear whether this is an inherent property of these cells or the result of disease progress, as suggested by D’Hooghe et al. (Reprod Sci, 2009). We therefore did not acknowledge this comment further in the discussion.

  • The pages in reference #2 “Sampson JA. Am J Pathol. 1927;3(2):93-110.143” should be “93-110.43.”

Changes were made accordingly.

  • The use of your reference #2 “Sampson JA. Am J Pathol. 1927;3(2):93-110.143” is incorrect for “backward menstrual flow.” Sampson’s 1927 Am J Pathol reference is to vascular dissemination not to “backward menstrual flow.” You appear to need his 1927 AJOG reference “Sampson, John A. (1927) Peritoneal endometriosis due to the menstrual dissemination of endometrial tissue into the peritoneal cavity. Am J Obstet Gynecol. 1927, 14(4):422-469, https://doi.org/10.1016/S0002-9378(15)30003-X.

 We thank the reviewer for pointing this out and have adjusted accordingly.

  • Highlighting Sampson in the title invites examination of his several theories. I agree with you that Sampson’s retrograde theory is likely the major cause of endometriosis. In addition, in the 1927 AJOG article, he suggests the use of both retrograde dissemination and differentiation of celomic epithelium to explain the diffuse locations. In 1927 (AJP and AJOG) he discussed vascular dissemination, retrograde flow, and celomic metaplasia. In 1921, he discussed dissemination from hemorrhagic (chocolate) cysts (Arch Surg (now JAMA Surgery). 1921, 3: 245-323, https://jamanetwork.com/journals/jamasurgery/fullarticle/536143), In 1925 (Am J Obstet Gynecol 1925, 10:649-664), he reviewed and supported remnants from Wolffian bodies transplantation endometriosis and extraperitoneal metastatic endometriosis. In 1925 he also discussed the possibility of developmentally misplaced endometrial tissue, but stated he had “never been able to appreciate it.” 1925 may be the first publication of the term “endometriosis.” Since “Sampson’s Theory” is not adequately defined, this can be confusing and since your title is theory dependent, consider discussing his use of multiple theories. Consider an explanation like “Sampson postulated several theories of endometriosis development. Retrograde menstruation is the most likely to explain the location of the majority of cases of intraabdominal endometriosis (Sampson 1927a (AJOG), Sampson 1927b (AJP)).” Also consider modifying the title to “Mimicking Sampson’s Retrograde Menstrual Theory in Rats: A New Rat Model for Ongoing Endometriosis-Associated Pain”

We completely agree with the reviewer and have clarified accordingly. As such, the title was adapted and more introduction to these theories were added in the introduction, see line 40-45: In the beginning of the 20th century, Sampson postulated several theories of endometriosis development, such as vascular dissemination, celomic metaplasia and retrograde menstruation [2, 3]. The latter is the most likely to explain the location of the majority of cases of intra-abdominal endometriosis [2, 3]

  • Sampson (AJOG 1927 stated that “both typical and atypical endometrial tissue was found and one could trace the transition of one type of lesion into the other.” Is your analysis of neuro-angiogenesis adequate to distinguish normal appearing transplanted endometrium from endometriosis as suggested by Koninckx’ endometriosis disease theory (Figure 2, Fertil Steril 2019, 111:327–40, DOI: https://doi.org/10.1016/j.fertnstert.2018.10.013)?

Both vimentin and cytokeratin positive cells were observed in the lesions, thus reflecting their endometrial character. As mentioned in a previous comment, in retrospect, the normal endometrial structure cannot be retrieved in histological analysis, indicating that this is rather abnormal endometrial-like tissue. Furthermore, they were scored on the occurrence of adhesion and neuro-angiogenesis, which was present in most of the lesions. However, the extent of neuro-angiogenesis was not linked with the “abnormality” of the endometriosis, as this was not the scope of the study.

  • Regarding “the oldest” hypothesis on line 39, the theories of 1927 were neither Sampson’s nor the literature’s oldest. Sampson had theories as least as early as 1921. An older review is in Russell (1899) who considered remnants of the germinal epithelium (Waldeyer, 1870), an extension of tubal epithelium (Marchand, 1879), a Wolffian body, and a Müller’s duct remnant. Russell WW. Aberrant portions of the Müllerian duct found in an ovary: Johns Hopkins Hospital Bulletin, 1899, 10:8-10 plus plates. If you search for that, you may link to Longo (Am J Obstet Gynecol 1979, 134(2):225-226) who reviewed Russell and other articles.

We acknowledge that poor wording has been used and have adapted the text accordingly, see lines 40-45 and, as mentioned in the previous comment.

Round 2

Reviewer 2 Report

  1. Thank you for your comprehensive revision and answers to my questions. The following are minor comments.
  2. Thank you for clarifying the lower increase in Figure 5A. Your wording is acceptable, but a decrease of an increase can be confusing. Consider using “less” rather than “decreased.” Thus line 230 would be “weight gain of ENDO recipient rat was significantly less over the course of 12 weeks compared to” and line 377-378 would be “However, the first assay showed that the body weight gain of the ENDO group was significantly less as soon as week 2 after ENDO induction”.
  3. Thank you for explaining “n.d.” Consider adding an explanation to the Figure 5. (B,C) legend about line 231-232 like “Some comparisons were not determined as there was a lack of measurements due to technical problems of the sensor.”
  4. On line 372, did you mean to keep “spontaneous” in that sentence?
  5. In Reference 3. “110 43.” is better as “110.43.”

Author Response

Reviewer #2:

Thank you for your comprehensive revision and answers to my questions. The following are minor comments.

Comments:

  • Thank you for clarifying the lower increase in Figure 5A. Your wording is acceptable, but a decrease of an increase can be confusing. Consider using “less” rather than “decreased.” Thus line 230 would be “weight gain of ENDO recipient rat was significantly less over the course of 12 weeks compared to” and line 377-378 would be “However, the first assay showed that the body weight gain of the ENDO group was significantly less as soon as week 2 after ENDO induction”.

We agree with the reviewer and the lines were adapted appropriately. Line 213-218: Two weeks after the endometriosis induction, weight gain of ENDO recipient rat was significantly less compared to SHAM group when normalized to baseline values (110.9 ± 0.7% vs 115.1 ± 0.9% (mean ± SEM)). This difference was maintained until the end of the experiment (122.9 ± 1.1% vs 130.1 ± 1.4% (mean ± SEM), ENDO vs SHAM) (Figure 5A).And line 378-380: However, the first assay showed that the body weight gain of the ENDO group was significantly less as soon as week 2 after ENDO induction compared to SHAM animals.

  • Thank you for explaining “n.d.” Consider adding an explanation to the Figure 5. (B,C) legend about line 231-232 like “Some comparisons were not determined as there was a lack of measurements due to technical problems of the sensor.”

A clarification for the lack of the data point was added to the manuscript, see line 232-233: Time point week 4 of the ENDO group was not determined due technical problems with the sensor.

  • On line 372, did you mean to keep “spontaneous” in that sentence?

To our knowledge, the sentence on line 371-373: Although hypersensitivity is seen in chronic pain patients, this symptom is often less burdensome in comparison to ongoing pain. Does not entail the word ‘spontaneous’ any more after the initial revision. If in the case that the version of the manuscript, which the reviewer received still, has this wording, we apologize and will remove this word appropriately. Through out the whole manuscript, the word spontaneous is now only used in the context of menstruation.

  • In Reference 3. “110 43.” is better as “110.43.”

Errors were adapted accordingly.